# Tunnel Configurations and Seismic Isolation Optimization in Underground Gravitational Wave Detectors

Florian Amann [1,2], Francesca Badaracco [3,*], Riccardo DeSalvo [4,5,6], Luca Naticchioni [7], Andrea Paoli [8], Luca Paoli [8], Paolo Ruggi [8] and Stefano Selleri [9,10]

1   Engineering Geology, RWTH Aachen, Lochnerstrasse 4-20, D-52064 Aachen, Germany
2   Fraunhofer Research Institution for Energy Infrastructure and Geothermal Systems IEG,
    Competence Center Geomechanics and Georisks, D-52064 Aachen, Germany
3   Centre for Cosmology, Particle Physics and Phenomenology, Université Catholique de Louvain,
    B-1348 Louvain-La-Neuve, Belgium
4   Department of Physics & Astronomy, University of Utah, 115 South 1400 East 0830,
    Salt-Lake-City, UT 84112, USA
5   Dipartimento di Ingegneria, Università del Sannio, C.so Garibaldi 107, 82100 Benevento, Italy
6   Riclab LLC, 1650 Casa Grande Street, Pasadena, CA 91104, USA
7   INFN-Sezione di Roma, 00185 Roma, Italy
8   EGO, European Gravitational Observatory, Via Edoardo Amaldi, 556021 Cascina, Italy
9   Department of Information Engineering, University of Florence, Via di S. Marta, 3, 50139 Firenze, Italy
10  INFN Sezione di Firenze-Urbino, Via G. Sansone, 1, 50019 Sesto Fiorentino, Italy
*   Correspondence: francesca.badaracco@uclouvain.be

**Abstract:** The Einstein Telescope will be a gravitational wave observatory comprising six nested detectors, three optimized to collect low-frequency signals, and three for high frequency. It will be built a few hundred meters under Earth's surface to reduce direct seismic and Newtonian noise. A critical issue with the Einstein Telescope design are the three corner stations, each hosting at least one sensitive component of all six detectors in the same hall. Maintenance, commissioning, and upgrade activities on a detector will cause interruptions of the operation of the other five, in some cases for years, thus greatly reducing the Einstein Telescope observational duty cycle. This paper proposes a new topology that moves the recombination and input–output optics of the Michelson interferometers, the top stages of the seismic attenuation chains and noise-inducing equipment in separate excavations far from the tunnels where the test masses reside. This separation takes advantage of the shielding properties of the rock mass to allow continuing detection with most detectors even during maintenance and upgrade of others. This configuration drastically improves the observatory's event detection efficiency. In addition, distributing the seismic attenuation chain components over multiple tunnel levels allows the use of effectively arbitrarily long seismic attenuation chains that relegate the seismic noise at frequencies farther from the present low-frequency noise budget, thus keeping the door open for future upgrades. Mechanical crowding around the test masses is eliminated allowing the use of smaller vacuum tanks and reduced cross section of excavations, which require less support measures.

**Keywords:** gravitational waves; Einstein Telescope; Newtonian noise; observatory; seismic attenuation; tunnel configuration; observational efficiency

## 1. Introduction

Gravitational waves (GW) are detected by the relative motion that they induce between suspended test masses separated by large distances. To detect that motion, four test masses are configured as the mirrors that compose two long Fabry–Perot interferometers that form the arms of a Michelson gravitational wave detector. The optical elements need to be extremely well isolated from vibrations (i.e., seismic induced) that otherwise would overwhelm the gravitational wave-induced motion.

### 1.1. Why Underground

The Einstein Telescope (ET) will be a European third-generation GW detector that will be built underground. It will be constituted by three nested GW interferometers in a triangular shape with sides of 10 km [1].

The ubiquitous seismic waves propagating with sub-micron amplitude through Earth's crust cause the so-called seismic noise, which is a mechanical movement that can be filtered away from a gravitational wave detector noise budget using sufficiently long and well-designed seismic attenuation chains [2]. Seismic waves also induce a much subtler effect that cannot be shielded. They cause tiny fluctuations of rock's density and position, which in turn generate tiny, local fluctuations of Earth's gravitational field that are seen by the detector as fluctuating space-time warps. This is called Newtonian noise [3]. Its effect on the test masses cannot be distinguished from the effect of a passing gravitational wave. The Newtonian noise acceleration depends linearly on the amplitude spectral density of the seismic displacement, which, at the frequencies of interest for ET, decays as $1/f^2$. Therefore, the Newtonian noise displacement amplitude spectral density will fall as $1/f^4$ [4] affecting the sensitivity below 30 Hz and intervening in the noise budget only at low frequency.

Gravitational wave detectors that are built underground are substantially less influenced by Newtonian noise. Yet, Newtonian noise will eventually limit the sensitivity at low frequency of any Earth based detector. At the surface, air density fluctuations due to infrasound pressure waves and wind turbulence contribute to Newtonian noise [5]. The air density fluctuations in the atmosphere are not too relevant for underground detectors that are sufficiently deep [6].

The large density difference between soil and air, the low elastic constant of soil and the larger displacement amplitude of both surface and body waves that increase their amplitude when surfacing cause large Newtonian noise on test masses hanging near the surface. Surface waves dominate the surface Newtonian noise spectrum, but their amplitude diminishes exponentially with depth and frequency [3]. This means that their contribution to Newtonian noise decreases exponentially with depth. Body waves are present at all depths. Even if their amplitude diminishes with the higher stiffness of deep rock, their Newtonian noise remain present at all depths. The depth chosen for the Einstein Telescope, about 300 m, was chosen as a tradeoff between diminishing returns and cost.

In theory, Newtonian noise could be estimated from a densely spaced network of seismometers surrounding the test masses and subtracted from the detector signal. This would require many high-precision seismometers optimally located in the rock surrounding the detector at distances comparable to the seismic wavelength generating the Newtonian noise, i.e., hundreds of meters. The method is further limited by the precision and the number of seismic sensors installed and by the mixing of shear and compressional waves that spoil the correlations between the inertial sensors used for the Newtonian noise estimation [7]. The quieter conditions inducing less Newtonian noise are the main rationale why future gravitational wave detectors aiming to detect gravitational waves down to 3 Hz [8] will be built underground. It will be shown that the proposed topology also allows almost optimal deployment of the inertial sensors required for an effective Newtonian noise subtraction.

All these improvements make ET a powerful and exciting instrument for gravitational wave astronomy.

An alternative approach, chosen for the Cosmic Explorer design [9], is to make a longer detector on the surface, diluting the fractional error of the larger surface noise with the larger amplitude of signals detected over longer distances. Subtraction of the much larger Newtonian noise present at the surface is possible but not sufficient to reach the desired low-frequency sensitivity of the Einstein Telescope. Underground locations are always advantageous to detect the lowest frequencies needed to acquire the signals from heavier black holes. Barring unforeseen developments to mitigate Newtonian noise, gravitational waves at frequencies close to or less than 1 Hz can only be detected with space probes such

as LISA [10] and DECIGO [11], or with Lunar gravitational wave antennas such as LGWA, GLOC or LSGA [12–14].

### 1.2. The ET Observatory Detection Efficiency

Gravitational waves happen all the time, and their signals will even overlap in the Einstein Telescope bandwidth. However, interesting events such as close-by neutron star inspirals or supernovae, that can give us detailed insight of astrophysics processes are rare [15]. The Einstein Telescope must be able to continuously observe gravitational waves to avoid missing rare events. In principle, it has sufficient redundancy (three nested detectors) for continuing observation even if a detector is out of service. This is a unique feature, but this redundancy is fragile. The catch is that in the present design, each detector requires commissioning, upgrade and maintenance activities stretching over years. As a result, even current single detector observatories such as Virgo and LIGO provide astronomical observation only for a small fraction of the time. The observational efficiency will be even worse, with six detectors needing attention or upgrades because, having at least one critical element of all six detectors in each of the three corner stations, activity on a single detector will impede observation with all the other five.

We present a tunneling configuration that physically separates all critical components to increase the observatory's effectiveness and allow uninterrupted astronomical observations.

### 1.3. The Xylophone Concept and Low-Frequency Sensitivity

It is foreseen that the Einstein Telescope triangle will host three pairs of interferometers [16]. Each pair will comprise a room-temperature detector specialized for higher frequency gravitational waves and a cryogenic detector optimized for low-frequency signals.

The room-temperature, high-frequency detector will run with more than a megawatt of standing optical power to reduce the shot noise and increase the sensitivity at high frequency. The cryogenic, low-frequency detector will use only kilowatts of stored optical power to minimize radiation pressure noise at low frequency and will require cryogenic, long, and soft suspensions as well as crystalline test masses to reduce thermal noise. In the noise budget of the present Einstein Telescope low-frequency design, the residual seismic noise, suspension thermal noise, Newtonian noise and quantum noise contribute at a comparable level to the low-frequency sensitivity limit. Suspension thermal noise can be pushed to lower frequency by longer, more flexible, and colder suspensions. Quantum noise can be mitigated by using heavier masses and quantum noise squeezing. Better sensors and technologies may be developed to subtract Newtonian noise. These improvements may happen during the expected 50-year facility lifetime. It is therefore important that the design of the seismic attenuators of the Einstein Telescope facility will not impose a limit to the low-frequency sensitivity, including for unforeseen future upgrades using techniques that are not yet anticipated.

The long isolation chains that can be implemented in the proposed configuration can push the seismic noise of the Einstein Telescope Low-Frequency detectors well below the other limiting factors, keeping an open door for further upgrades. Perhaps more importantly, the long chains may ease some of the control noise issues. The proposed underground configuration also allows placement of inertial sensors in spherical patterns for optimal Newtonian noise subtraction.

## 2. Structure of Present and Future Detectors

Terrestrial gravitational wave detectors are kilometer-scale Michelson interferometers with arms equipped with Fabry–Perot cavities to extend their effective optical length. The Fabry–Perot mirrors are heavy test masses suspended from threads, so that they can be freely accelerated along the beamline by the space-time fluctuations caused by passing gravitational waves. Present surface detectors (Virgo, LIGO, KAGRA [17–19]) have arms 3 to 4 km long, while future detectors currently being designed will be three to ten times longer. In the current surface L-shaped detectors, input test masses, recombination and

input–output optics are housed in a single large building. The present design of the Einstein Telescope has three 10 km-long tunnels oriented at 120° from each other. The six nested detectors are intended to record signals from both polarizations of gravitational waves. They intersect in three large vertex halls, each containing eight of the twenty-four test masses, including at least one of all six detectors, as well as the input/output optics of two Michelson interferometers and their ancillary equipment. This configuration has several drawbacks including excessive crowding and mixing of optical components and seismic isolation systems, that require quiet conditions, with ancillary equipment, such as cryogenic chillers that cause reverberating acoustic noise, which is known to inject noise into the interferometer. The worst concern is that access for maintenance or upgrade on a single detector will affect, and in most cases impede, the operation of all others, thus reducing or interrupting astronomical observations every time an access is made.

## 2.1. Underground Stability Constraints

The Einstein Telescope will be built at about three hundred meters below the surface. Strong and massive, moderately jointed (i.e., unfractured or slightly fractures) rock mass conditions, dry or with a low hydraulic conductivity are assumed. Underground construction causes a stress redistribution around the excavation that may exceed the rock strength or cause rock creep [20]. In tunnel designs with many adjacent large excavations, the redistributed stresses of individual excavations will superimpose and intensify. It is therefore important to allow for rock walls wider than the excavated volumes between large voids, such as at the intersection of the halls at the corner points to avoid failures started by blast damage or superimposed stress states.

The proposed design, sketched in Figure 1, is only indicative, the lengths and excavation shapes must be optimized to satisfy science requirements and avoid all structural weakness. Large excavations often suffer from structural controlled instabilities, e.g., rock wedges, that are formed by intersecting discontinuities such as joints or small fault planes, which require heavy rock support, e.g., long and heavy rock bolts or anchors, thick reinforced concrete lining, and more. The stability issues at the target depth of the Einstein Telescope are mostly relevant for the large caverns in the corner points, where the size of the excavation damage zone or structural controlled rock wedges could also be large. The instability is fully mitigated by the reduced size and rounded shapes, i.e., curved side walls instead of vertical side walls. Detailed size and shape adjustments can be performed in an early design stage by optimizing the distribution of the detector components and their mode of access through the rock mass once the rock mass conditions are known through a site investigation program.

## 2.2. Other Underground Constraints

In addition to stability considerations, the corner stations host access tunnels or shafts with constraints that need to be considered. The ET facility is expected to last 50 years. The LIGO Livingston observatory experience teaches that in an environment with high humidity, after only 20 years, corrosion can degrade the vacuum works and cause leaks even in pipes built with stainless steel. Underground water and airborne agents are usually more aggressive than at the surface. Water ingress must therefore be avoided either by an optimal positioning of the facility in dry rock with low permeability or by systematic grouting, sealant injections and surface lining to reduce the rock mass hydraulic conductivity and eliminate accumulating or steady-state inflow rates. Any water must be immediately collected through a drainage pipe system and routed out. Available data indicates that the Sos Enattos site is likely to satisfy this important requirement. The local mine drainage flow is of the order of one liter per second from its ~50 km of galleries and the electrical resistivity tomography of the site shows the absence of significant groundwater, due to the low porosity of the rock [21].

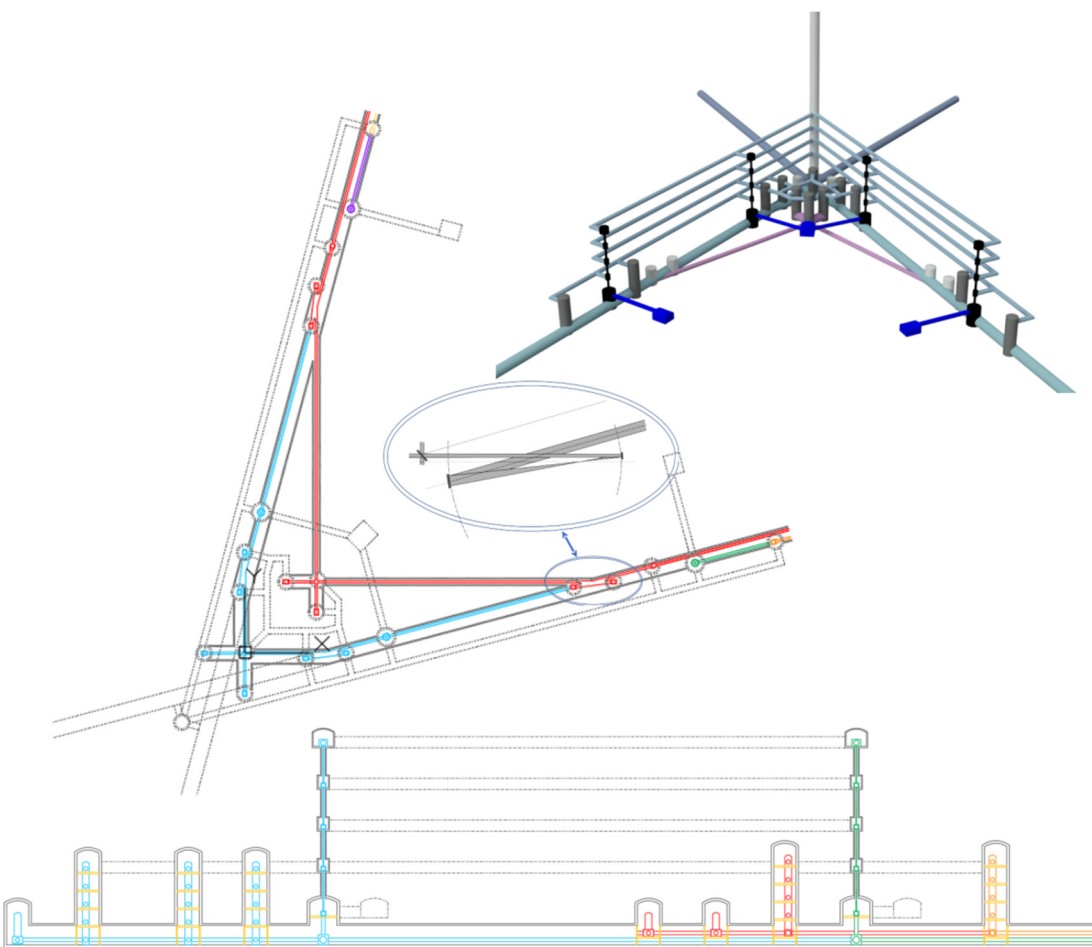

**Figure 1.** (**Top left**) Configuration for separate extraction of the two interferometers from the main tunnel and a possible structure for access tunnels. The insert is a zoom onto the beam-reducing telescopes that extract the Michelson beams from the main tunnel. The room temperature interferometer, which is located above the cold one, (red) is extracted first, thus leaving more space for the cryostats of the low-frequency detector (blue). Input and output optics (not detailed here) are housed together with the beam splitter in separate L-shaped excavations of reduced volume. (**Bottom**) Elevation view of vertical structures with different kinds of seismic attenuation chains. All access tunnels (behind or in front of the tunnel vertical section) are represented with dotted lines. The 3D insert on the top right corner is a sketch provided to guide the eye, and an interactive 3D version is provided in Figure A2 of the online-only Appendix A.

### 2.3. The Proposed Tunnel Configuration

The tunnel configuration sketched in Figure 1 takes best advantage from the underground environment for building gravitational wave observatories. It consists of several small and interconnected stable excavations. Angled, beam-reducing telescopes positioned upstream of the input test masses produce extracted beams that propagate at a 15° angle away from the Fabry–Perot direction, so that the recombination on the beam splitter happens at the optimal 90°. A suitable separation along the tunnel of the input test masses of the two Michelson interferometers causes the extracted beams to cross in separate excavations that house the beam-splitter and input/output optics [22].

The fundamental limitations of low-frequency sensitivity of gravitational waves, i.e., quantum noise, radiation pressure, and suspension thermal noise are not addressed here. We only note that mitigation of suspension thermal noise requires substantial vertical space above the test masses, which is made available by this configuration. The proposed scheme is also suitable for upgrades to heavier masses to reduce the quantum back-action noise.

### 2.4. Present Seismic Attenuation Systems

Seismic noise attenuation comes from the natural properties of a pendulum, which attenuates horizontal vibration transmission with a $1/(f^2 - f_0{}^2)$ function that provides a cutoff starting above the pendulum resonant frequency $f_0 = \sqrt{(g/l)}/(2\pi)$ where $l$ is the pendulum length and $g$ is the gravitational acceleration. The mechanical attenuation is provided by a chain of pendulum wires alternated to massive vertical attenuation filters (discussed in Section 3.3) acting as pendulum mass. Each stage contributes a $1/(f^2 - f_0{}^2)$ attenuation starting from its corresponding resonant frequency. With $n$ stages a $1/(f^2 - f_0{}^2)^n$ attenuation power is achieved [23]. Therefore, seismic attenuation performance on the low-frequency side is limited by the length of the pendulums used, which are limited by the height available above the test masses.

The overall length of the seismic attenuation chains of present detectors ranges from a couple of meters in LIGO [17], to seven meters, in Virgo [18]. Most present detectors (except for KAGRA) rely on support structures extending up from the tunnel floor encroaching in the volume around the test masses. Tall structures need to be very stiff to avoid amplifying ground motion and their height is limited by the hall that hosts them. Individual pendulums in present detectors are limited to lengths of the order of a meter, with resonant frequencies (start of passive attenuation) distributed around 0.5 Hz. This is adequate for a detection threshold above 10 Hz but not for the lower detection frequency of the Einstein Telescope.

### 2.5. Present Design of Einstein Telescope Seismic Attenuation System

The Einstein Telescope aims to be sensitive below 10 Hz, thus requiring the start of the $1/(f^2 - f_0{}^2)$ roll offs at lower frequencies than present detectors, and therefore longer pendulums. The attenuation chain length in the current Einstein Telescope design is limited by the foreseen excavation ceiling height to 17 m tall towers [8]. Each tower contains six stages of pendulums with less than 3 m long suspension wires. The resonant frequencies of 3 m long pendula are close to 0.3 Hz (0.49 Hz in Virgo 1 m pendula). With these limitations in vertical size, seismic noise remains a small but non-negligible source of noise at the lowest frequencies, which may become a limiting factor if new ways are found to further depress quantum, suspension thermal and Newtonian noise.

### 2.6. The Proposed Seismic Attenuation Structures

The two taller structures in the lower panel of Figure 1 are stacks of excavations (alcoves) connected by a vertical borehole. Each alcove harbors one of the vertical attenuation filters separated by pendulums longer than 10 m. They provide isolation for the most demanding cryogenic test masses of the low-frequency interferometers. The dotted side excavations near the base of these stack (dark blue in the 3D insert) house the noisy cryogenic chillers. Separate excavations are needed to isolate the vibration noise caused by the more than 150 kW of cryo-chiller power needed to cool the test masses and long vibration-isolated cryogenic shroud and baffles. These baffles extend for more than 50 m and have two requirements. They are needed to intercept all possible trajectories of water molecules that may otherwise reach and deposit on the mirror surfaces and must be vibration isolated to neutralize the effects of residual scattered light. The cooling power is delivered at an appropriate height along the seismic attenuation chains, with an appropriate separation from the test masses.

Any other noisy equipment can be housed in similar alcoves. The access tunnels can be configured to reroute most of the main tunnel ventilation away from the test masses and recombination optics, where turbulence may cause atmospheric Newtonian and acoustic noise.

The five intermediate-length towers, also shown in Figure 1, are for the room-temperature interferometer test masses and other main optical elements. They may be of different kinds and heights, depending on specific requirements. The three smallest structures contain seismic isolation for optical benches and other less demanding optics. An important point is that all seismic attenuation chains can be anchored to the rock above to avoid encroachment

of support structures around the optical elements. The inverted pendulum and top filter at the head of the long chains sit directly on the rock floor in the top alcove.

It is important to note that alcoves with narrow and sealable access tunnels proved very effective to isolate from ambient noise or vibrations induced by ventilation or other machinery. When installing seismometers in the Homestake mine [24], it was observed that the instruments installed in similar alcoves had the lowest noise floor.

The elimination of structural encroachment around the test masses is especially valuable in the locations where the beams of the high-frequency interferometer peel off from the main tunnel and to make space for the cryostats. The test masses can be simply housed along the main tunnels with little or no local tunnel enlargement.

Other advantages provided by the proposed configuration are detailed in the following sections.

We stress that the sketch of Figure 1 is only conceptual, and reasonable dimensions are suggested but not optimized. The exact topology, height, load, or size of attenuation stages for different optical elements are not yet determined. The vibration isolation solutions outlined can be scaled to meet the different requirements of various optical elements, while easy access to all attenuation chain elements can be maintained. Similarly, no effort is made here to decide excavation shapes and sizes, or to decide spacing along the beams to eliminate interference between different detectors. This will be the object of a complex tradeoff study involving mining engineers and physicists, with the aim to minimize costs, maintain safety and satisfy all scientific requirements. A rough cost comparison between the ET baseline design and the one proposed here was made [25]. The larger excavation volume of the first balance the greater complexity of the second. Within the evaluation error margin, no significant cost difference was found.

### 3. Advantages of the Proposed Configuration

#### 3.1. Segregating the Critical Components of Different Detectors in Separate Caverns

Decoupling the maintenance, upgrades and/or commissioning of different detectors in the Einstein Telescope is of crucial importance because, it allows researchers to work on one detector without affecting the operation of the others and to maintain continuous observation mode.

The separation of the recombination and input–output optics of different detectors in individual excavations and their separation from the test masses is discussed in [22] and illustrated in Figure 1. The key point is that just a few meters of rock provide much more isolation of what exists between the interferometers and their control rooms in present surface detectors. Similarly, access to the main tunnels housing the test masses may not affect the operations because the support points of the seismic attenuation chains from which vibrations may be injected reside on hard rock at higher tunnel levels and, therefore, are insensitive to activity in the main tunnel. Working on a detector can be expected not to affect the operation of the other five.

#### 3.2. Low-Frequency Seismic Noise and Reduction in Out-of-Band r.m.s. Motion

There are serious concerns about the interferometer control noise. Forces are needed to maintain the lock of the interferometer with sufficient authority to compensate for the residual motion of the suspension chains below the gravitational wave detection band. These out-of-band control forces can generate up-conversion and inject control noise in the detection band [26]. There is substantial advantage in lowering the isolation system resonant frequencies and in reducing the amplitude of the pendula residual motion. The longer oscillation periods of a longer pendulum allow more time for a force to act. Smaller control forces are required to control smaller amplitude motion. In both cases, less control forces are required, and this results in less control noise.

### 3.3. The Attenuation Chains as Seismic Sensors

Sensing the changing distance of the seismic attenuation filters with respect to the surrounding rock at various heights along the attenuation chains provides a measurement of the linear and tilt movements for active damping of the pendulum modes. It is also important to reduce the movements induced on the test masses by the micro-seismic peak [27,28]. In this regard, it should be noted that after acquisition of Fabry–Perot lock, each pair of test masses effectively form a single, rigid inertial mass hanging from two widely separated points subjected to the seismic noise of two vertices of the triangle. After the Michelson lock acquisition, the four test masses of each interferometer form an even larger composite inertial mass with suspension points distributed in all three facility vertices. Optical levers, augmented with fringe counting, provide length sensing between each test mass and the surrounding rock. In the Einstein Telescope triangular topology, the six locked detectors provide multiple independent measurements of the low-frequency seismic motion with two different suspension frequency tunings and three orientations. These redundant signals can be combined to form synthetic seismometer measurements of the low-frequency seism at the three vertices with a noise floor substantially below what can be achieved with conventional sensors. These signals can be used to suppress the amplitude of low-frequency motion caused by micro-seismic noise. All of this contributes to reduce the residual motion of the test masses and ultimately control noise.

### 3.4. Multiple Tunnel Layers for Lower-Frequency Seismic Noise Suppression

The multiple level topology provides effectively unlimited vertical space and pendulum lengths. The start of the $1/f^2$ roll off at lower frequency shifts the seismic noise farther from the Einstein Telescope detection frequency band. Consider, for example, filters housed in 4 m-tall excavations vertically separated by 8 m of rock. The resulting ~12 m separation between filters is four times that in the current Einstein Telescope design and produces pendulum resonances below 0.15 Hz. This arrangement, assuming the same number of filters, would cut in half the starting frequency of the seismic attenuation roll off. This of course works only if that frequency can be matched by vertical attenuation filters [28,29] also tuned at or below 0.15 Hz (see Section 5 for detail).

The attenuation performance of a 60 m long chain is compared in Figure 2 with that of the 17 m chain of the Einstein Telescope baseline design. The expected start of the attenuation curve at half the frequency is visible in the simulation at ~0.55 Hz. The structure above 0.55 Hz is contributed by suspensions that are not yet re-optimized. The displacement noise is compared in Figure 3 by folding the attenuation transfer function of Figure 2 with a typical underground seismic spectral noise (data from [30]). Additional simulations show that with longer suspensions, made also possible by the proposed topology, the full gain from the longer wires can be recovered.

### 3.5. Separating the Top of the Seismic Isolation Chain from the Test Mass Level

The scheme for seismic attenuation illustrated in Figure 1 envisions vertical stacking of small excavations connected with boreholes for the pendulum wires. The only connections to ground are through the pre-isolators, which are relegated to the top level, and through the cryogenic heat links [31]. The cryogenic heat links can be positioned at a suitable intermediate level of the attenuation chain to screen the test masses from residual mechanical noise from the chillers.

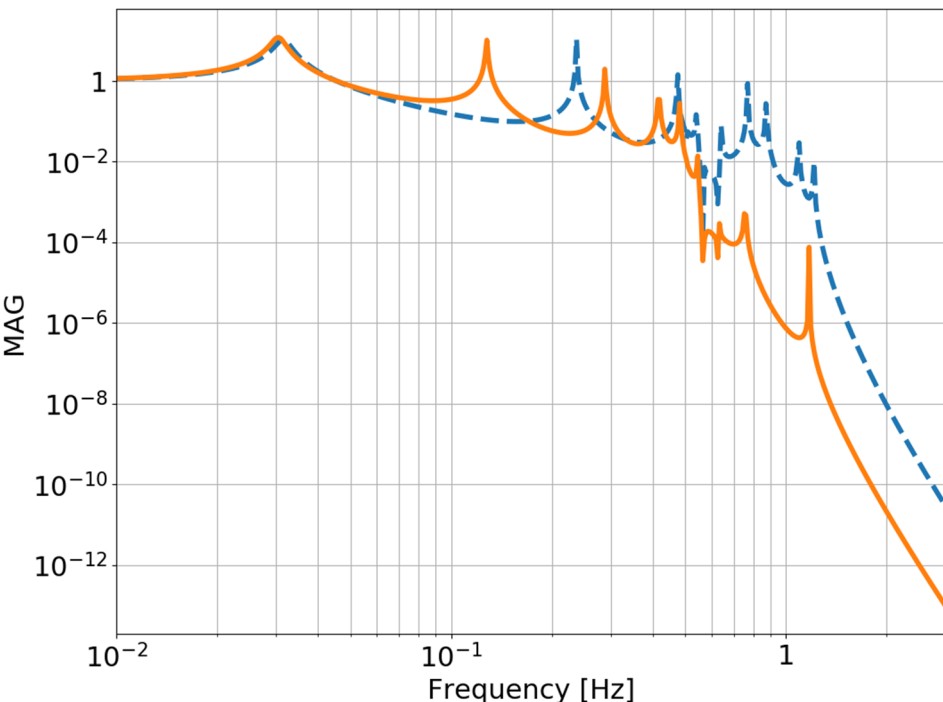

**Figure 2.** Comparison between the Einstein Telescope baseline seismic attenuation scheme with 3 m separation between filters (orange, solid) and the same configuration with 12 m (blue, dashed) of separation between filters. The mirror suspensions are the same in the two simulations, without re-optimization for lower frequency performance.

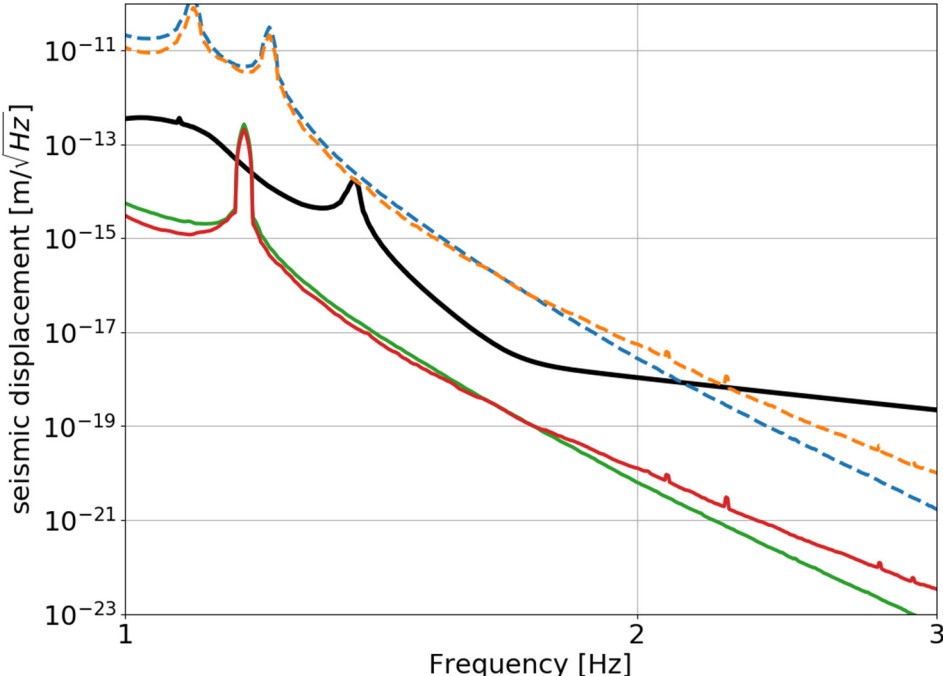

**Figure 3.** Comparison of the expected displacement noise after convoluting the transfer function with the measured seismic noise in Sos Enattos (blue and green) and Terziet (orange and red). The seismic data are taken to be the square root of the squared sum of the 90th percentile of the north and the east channels. The dashed lines refer to the 17 m suspensions and the solid lines to the 60 m suspensions. The optical length sensing is expected to have a sensitivity around 10–18 m/$\sqrt{Hz}$ in this frequency range. The black line is the current design sensitivity curve of the Einstein Telescope.

### 3.6. Smaller Vacuum Chambers

In Virgo and LIGO, the seismic isolation systems had to be contained and supported from the ground by bulky vacuum chambers requiring human access for installation and maintenance.

Having relocated the seismic attenuation away from the main tunnel level, much smaller vacuum chambers can be used, including for the test masses. Similarly, moving the cryogenics in separate excavations, and injecting the heat links higher up in the attenuation chain allows the use of smaller gauge tunnels. While the larger payload integration in the Einstein Telescope may still require access from below, as in Virgo, other operations of maintenance and tuning of the system may be possible from side access. Much cleaner working conditions than for the test masses of the present detectors can be obtained if the operator can work from the side of the vacuum chamber while the optics are continuously protected by a flow of clean air provided from above. This kind of side access has been successfully used in Japan since the TAMA detector [32]. The scheme illustrated in Figure 4 was designed for beam size reducing telescopes in KAGRA [33] that use recycled, large initial LIGO test masses. The same side access configuration can be adapted for the larger and cryogenic optics of the Einstein Telescope.

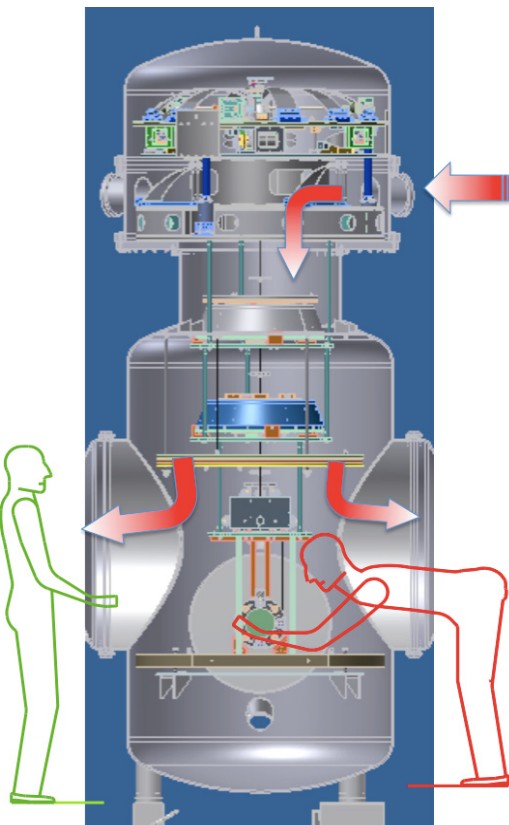

**Figure 4.** Ergonomic access to large optics in a reduced-size, side-access vacuum tank. The operators remain outside the vacuum chamber, thus reducing chances of introducing pollutants. The red arrows indicate a clean air flow. Having removed the encumbrances from the seismic attenuation structure, the same scheme can be applied to larger size optics, including cryostats equipped with removable lateral thermal shield panels, such as those implemented in the CLIO and KAGRA cryostats.

### 3.7. Access to Suspension Elements of the Cold and Warm Interferometers

Access to upper elements of a seismic attenuation chain has always been a laborious enterprise, requiring removal of the vacuum chambers and involving tall and complex support structures that extend along the full chain's height. These structures recur in the

wide excavations proposed for the Einstein Telescope and occupy a significant fraction of the hall volume. Installation and upgrades pose significant technical challenges and risks.

In the configuration proposed here, side-access vacuum chambers in vertically separated alcoves provide easy and safe access for installation, tuning and upgrades of the long seismic attenuation chains.

### 3.8. Shorter Seismic Isolation Chains

Seismic isolation for less demanding and auxiliary optics can be satisfied with more compact chains such as those used for the main mirrors in KAGRA, as illustrated in Appendix A Figure A1 (left). An example of the complex mechanical structures required to mitigate the amplification of ground tilt on raised structures is illustrated in Appendix A Figure A1 (right). This complexity and space encumbrance can be avoided even for short chains by attaching the top stage of their seismic attenuation chain to the rock above. The shorter chains leave no place for rock-separated alcoves, and rock-anchored platforms in wells will fulfill the same function of alcoves to provide easy access.

### 3.9. Segregating Noisy Equipment

The cryogenic chillers for the 12 cryogenic test masses require a power of the order of 150 kW [34]. Chillers are notoriously noisy, especially at low frequency. To isolate the vibrations that they generate, it is necessary to house them in separate excavations, with anchoring to rock designed to filter out compressor vibrations and heat links running through boreholes. Several meters of rock efficiently shield the vibrations while the machinery can remain continuously accessible for maintenance via separate tunnels. It should be noted that the longer lengths involved may require replacing the external metallic heat links with liquid helium lines, which further reduce transmission of vibrations.

It is worth mentioning that water cooling will be necessary to evacuate the power consumed by the chillers, as well as for any other instruments requiring significant power. Air cooling cannot be relied on because fast air flows through the tunnels, producing unacceptable levels of induced noise, including Newtonian noise.

## 4. High-Frequency Detectors and Shorter Chains

The Einstein Telescope High-Frequency detectors [16] feature less demanding attenuation chains with shorter pendulums. The stacked alcove configuration becomes unnecessary. These attenuation chain components can reside in rise-bore shafts extending above the main beam tunnel. The pre-isolator would rest on short beams anchored to the rock. The number, separation and size of filters needed would depend on the requirement of each specific optical element. Cylindrical vacuum chambers with side access flanges and balconies, all supported on the rock, provide ergonomic access to the chain elements for maintenance and tuning.

## 5. More on Seismic Attenuation

### 5.1. Vertical Attenuation Filters

The attenuation performance in the horizontal plane is spoiled if not accompanied, at every stage, by matching vertical attenuation. Because of unavoidable mechanical imperfections and of the effects of Earth's sphericity, vertical noise leaks into the horizontal plane at each step of an attenuation chain [2]. Suitable vertical attenuation can be achieved using modular Geometric Anti-Spring filters in appropriate number and size to match the requirement of each individual optical element. The Geometric Anti-Spring filter is a mature technology used in Virgo upgrades [35], HAM-SAS [36], TAMA [37], KAGRA and scientific and commercial platforms. They are sophisticated but simple mechanical oscillators that can be tuned to low resonances but were never required to operate at or above 0.15 Hz. Examples of two kinds of filters that can be re-sized to satisfy all the Einstein Telescope requirements are illustrated in Figure 5. The low-frequency operation (0.12 Hz) shown in Figure 6 was achieved

for the top Geometric Anti-Spring filters of KAGRA. This tuning already matches the resonant frequency of the longest pendula envisaged in this paper.

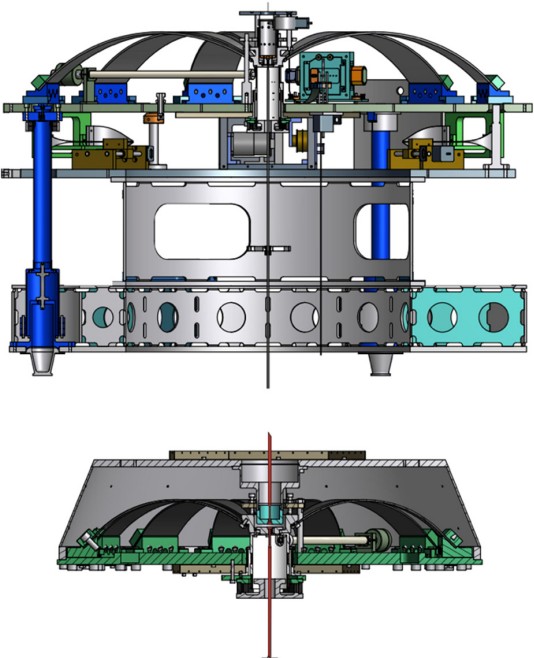

**Figure 5.** A pre-isolator (**top**) and of a standard filter (**bottom**) designed and built for KAGRA. The pre-isolator is composed by a short, inverted pendulum footed directly on the bedrock in an alcove at the top. It supports a large Geometric Anti-Spring filter that in its turn suspends a chain of standard filters. The standard filter is similar but smaller. Both can be scaled to the heavier payloads of the Einstein Telescope.

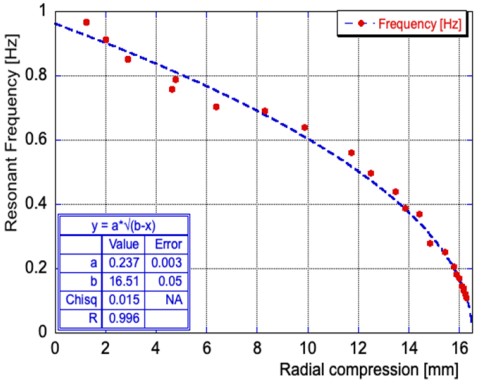
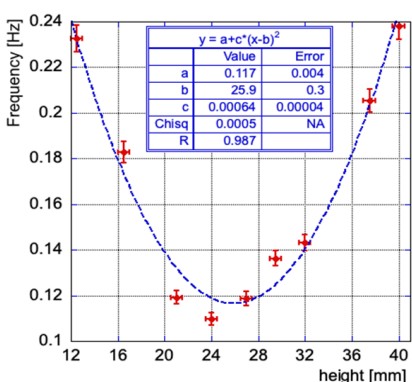

**Figure 6.** Tuning of a KAGRA pre-isolator Geometric Anti-Spring filter. (**Left**) Frequency tuning of the resonant frequency obtained by radial compression of the cantilever blades forming the Geometric Anti-Spring mechanism [38]. (**Right**) Dependence of the resonant frequency from the working point. The working point can be fine-tuned in situ via controls of the temperature of the vacuum tank.

A caveat is that low resonant frequency is achievable only over a very narrow dynamical range, as evident in the right panel of Figure 6. If the resonant frequency is tuned to lower values, the frequency vs. height curve becomes progressively narrower. This tuning requires matching the spring strength to the payload weight at $<10^{-4}$ level, which can be achieved with thermal control of individual filters. The Young modulus of steel changes by $\sim 3 \times 10^{-4}$/K with temperature. The required load matching can be achieved with a thermal control of a tenth of a degree, which is easily achievable in a vacuum tank. A range of several degrees is needed while up to 100 K of heating is allowable without inducing creep on properly treated maraging springs [39].

### 5.2. Pre-Isolator Short Legs

A pre-isolation filter is foreseen at the top of each optical chain. It is composed by a short, inverted pendulum supporting a Geometric Anti-Spring filter. Short inverted-pendulums have the same attenuation power of longer ones without being affected by the low-frequency internal resonances (banana modes) of longer legs. Long legs may weigh more than 100 kg while short ones supporting the same load weigh less than a kg. Therefore, their batting center and the recoil of their internal modes affects much less the heavy top filter. The counterweights for batting center compensation are less critical, and in some cases unnecessary [40]. By sitting directly on hard rock, the pre-isolators do not suffer from the amplification of ground tilt affecting tall structures [41,42], this is an invaluable advantage. The simple coupling of an inverted pendulum and a Geometric Anti-Spring filter [43,44] provides a passive attenuation power equivalent to that of the entire Advanced LIGO active isolation without its control complexity [45]. While fundamentally a passive element, the inverted pendulum and top filter combination is also an ideal platform for the active reduction in low-frequency seismic noise with inertial sensor signals [46,47] or to apply feed-forward from other signals [48,49].

### 5.3. Test Mass Suspensions

The test masses in present and future room temperature detectors are suspended using fibers made of fused silica [50]. This is also the solution chosen for the high-frequency detector of the Einstein Telescope. Silicon, sapphire, or other crystal rods will be needed to suspend and cool test masses in low-frequency cryogenic detectors [51,52]. In the Einstein Telescope, the suspensions will need to be longer and softer than in present detectors to reduce the suspension thermal noise that dominates at low frequency; yet, they must have sufficient thermal conductivity to keep the mirrors at cryogenic temperatures. The design of the mirror suspension solutions with large compliance and thermal conductivity are considered elsewhere [53]. The proposed configuration provides the vertical height needed for suspension thermal noise mitigation.

## 6. Flexibility Gains and Limitations

The large halls in the present Einstein Telescope design are intended to provide flexibility for future improvements of the facility. It has been argued that positioning seismic attenuation chains in wells would impede length tuning or reconfiguration of the optical cavities. While it is true that large changes in the Fabry–Perot lengths will not be possible without re-shaping boreholes and excavations, there is little reason to do it, and small length changes for tuning purposes remain possible within the diameter of the connecting boreholes. It should be considered that the input/output optics located in individual recombination halls will be easier to reconfigure because there is no interference with other detectors.

## 7. Radial Boreholes and Newtonian Noise Subtraction Sensors

Effective Newtonian noise subtraction techniques require numerous high-sensitivity sensors to instrument the rock mass surrounding the test masses. These sensors must be distributed in a pattern with a size comparable with the wavelength of the seismic waves of interest. At 2 Hz, with a seismic speed of 5 km per second, the characteristic dimension is of the order of a kilometer. This is a large and expensive endeavor involving tens of kilometers of boreholes that may be justified only when the detector has reached sufficient sensitivity at low frequency. Experience shows that it takes years to build a gravitational wave detector and even more to reach the stage when Newtonian noise may become a limitation. It is thus necessary that the Newtonian noise mitigation can be staged and/or implemented at a later time, when better understanding of the requirements is gained.

The required Newtonian noise cancellation system can be achieved with maximized effectiveness, minimized borehole length, and minimized cost by directional drilling of boreholes radiating from tunnels located near the three corners. Directional boring tech-

niques developed for oil extraction and ore exploration [54] allow an optimal and easily expandable sensor geometry. Boreholes radiating from near the corner stations are sketched (in a plane) in Figure 7. This geometry minimizes the drilling length and allows centralized signal collection. Implementation will require custom triaxial inertial sensors with remote anchoring/release and alignment mechanisms. Boreholes hundreds of meters in length can be drilled from one or two dedicated tunnels to instrument the volume surrounding the corner stations. Each borehole will host many sensors spread out at suitable distances to satisfy the optimal sensor distribution requirements.

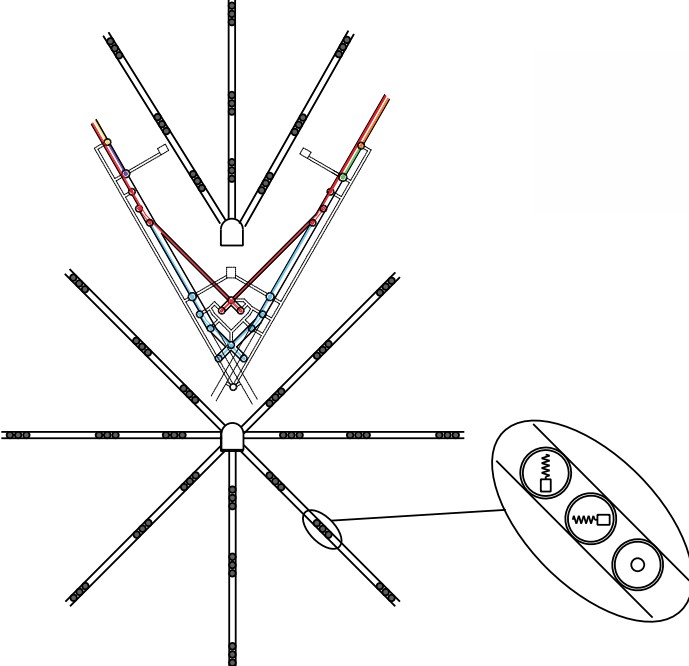

**Figure 7.** Schematic view (not to scale) of radial boreholes and inertial sensor positioning around the corner station for optimal Newtonian noise subtraction. A tunnel in the front and one on the back of the corner station are hypothesized here. The angular density per steradian, the length of the boreholes and the longitudinal positioning of the sensors are chosen to optimize the sensor density for Newtonian noise subtraction while minimizing the borehole length. A crawler positions, levels and aligns the sensors at the desired distance along the borehole (see text for details).

The mono-axial sensing heads inside modern low-frequency seismometers are small and require only small changes to fit within standard diameter boreholes, i.e., 130–170 mm but precision self-alignment capability will be necessary. The installation must be performed remotely by crawlers capable of installing and aligning or retrieving the sensors. An installation procedure might be as follows. A mono-axial sensor is brought in place by a cable-controlled crawler. The sensor's outer housing is equipped with spring-loaded expansion brakes for rigid connection to the borehole walls while a spherical bushing allows smooth rotation of the sensor head. Both the brakes and the bushing are of the normally locked type. They are released as needed with compressed air provided by the crawler. After positioning, the crawler locks the brakes to the walls, orients the sensor to level and align it, then locks the rotation. Finally, the crawler returns to fetch the next sensor that is connected to the previous one before starting its alignment procedure. The sensors are daisy chained along the borehole and installed in groups of three to get three-axial sensing while the spacing between the triplets is determined by the sensing requirements.

Boring produces vibrations. The tunnels hosting the drilling equipment must be at a distance, to be determined experimentally, from the detectors that the seismic attenuation chain can neutralize them.

Drilling vibration may still affect the length sensing of the interferometers through other channels such as scattered light, possible up-conversions of control signals, and even Newtonian noise. The radiated vibration can be monitored with geophones placed near the source. Drilling can be regarded as a "known" point-like source of seismic noise. If any effect on sensitivity is observed during drilling operations, modern signal correlation techniques may identify, quantify, and possibly localize channels of length sensing noise injection. Drilling may therefore be an effective diagnostic tool to identify and eliminate, or exclude noise injection channels that otherwise may remain poorly understood noise intrusion paths limiting the sensitivity.

## 8. Conclusions

The first and foremost gain from the tunnel topology proposed here is the improvement of the observatory astronomical observation efficiency. The physical separation of the test masses, the recombination optics of different detectors and the support point of the isolation chains, as well as the relegation of noisy equipment in separate and sealed excavations, will allow continuous astronomical observation mode in most detectors even while work is performed on others for staged installation, maintenance, or upgrades. Gaps in gravitational wave observations and the risk of missing a rare, potentially multiple-messenger event such as a close-by supernova or a neutron star merger are eliminated.

The second advantage is that arbitrarily long attenuation chains built into the rock environment push the seismic noise farther from the detector noise budget and keep the door open for future sensitivity improvements. Perhaps more importantly, lower frequency attenuation and smaller residual motion reduce the force required for controls and the chance that control noise may end up limiting the detection sensitivity.

The cost of boring and of high-quality inertial sensing instruments is relevant, and the quality of sensors continuously improves. It will take years before the Newtonian noise represents a limit to sensitivity. Therefore, it is probably wise to delay implementation of instruments for full-fledged Newtonian noise subtraction. The proposed tunnel topology is suitable for staging without interrupting astronomical observations.

**Author Contributions:** F.A. is an expert of mining and geology, contributed mining and geologic considerations, A.P. and L.P. are mechanical engineers at EGO, contributed the drawing describing the topology and design of the facility, P.R. is one of the Virgo Observatory oprators and contributed the instrument and controls point of view, as well as performance evaluations using standard Virgo tools, S.S. is expert of optics design and contributed the optics point of view, F.B., R.D. and L.N. are Gravitational Wave detection scientists that served in all present Gravitational Wave observatories. F.B. focussed on the Newtonian Noise issues and served as the main editor of the paper, L.N. on the geophysics issues, R.D. originated the core idea and assembled the group of expert to cover all the elements needed to flesh out the concept. All authors have read and agreed to the published version of the manuscript.

**Funding:** This research received no external funding.

**Data Availability Statement:** The original contributions presented in the study are included in the article, and further inquiries can be directed to the corresponding author.

**Acknowledgments:** The authors would like to thank the reviewers and editors for improving this manuscript. The authors are grateful to the ET Sardinia host team, to the Istituto Nazionale di Fisica Nucleare (INFN) and the Italian Ministry of the Education, University and Research (MIUR), under the agreement 0021983-06/03/2018, for the borehole seismic data used with the suspension transfer functions studied in this work. RDS likes to remember the many discussions with his late father Francesco, mining engineer and mine director, who contributed many of the ideas presented in this paper. The figures representing examples of seismic attenuation components were originally provided by RDS for KAGRA. We would like to thank Alessandro Bertolini, Aniello Grado, Francesco Fidecaro, Giacomo Oggiano, and Joris van Heijningen for useful criticism and discussions, and Gianni Gennaro for the KAGRA graphics.

**Conflicts of Interest:** The authors declare no conflict of interest.

**Appendix A Online Only**

In this section, we present:

- Examples of a typical seismic attenuation chain and illustration of the kind of external support structures that may become necessary if the seismic attenuation chains are not connected directly to hard rock;
- An interactive illustration of a tunnel topology satisfying the requirements of the Einstein Telescope detectors.

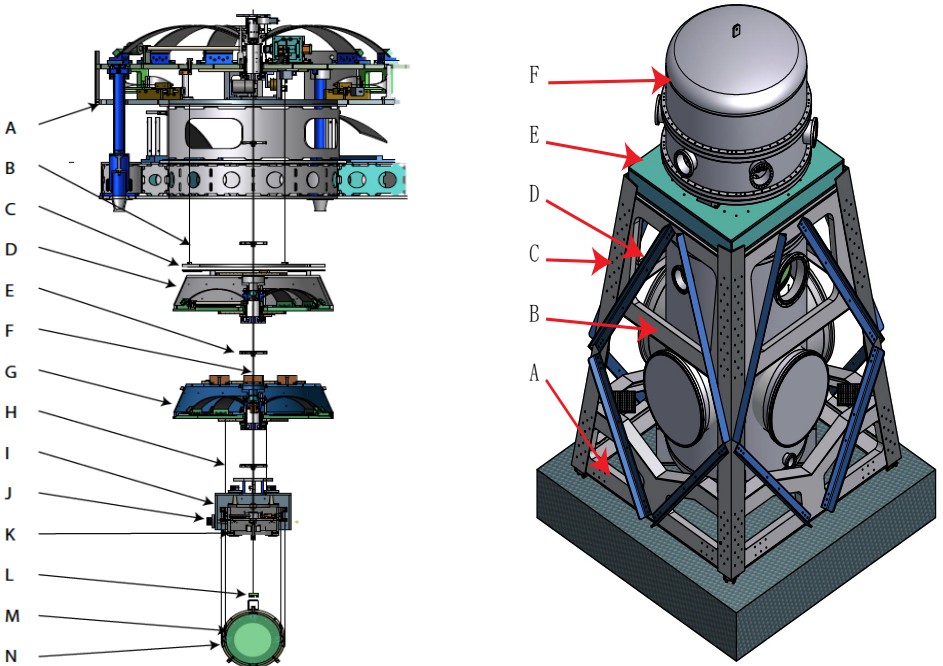

**Figure A1.** (**Left**) Example of a typical seismic attenuation chain; length of wires, size and number of filters would depend on the mass and requirements of the optical payload. A: Pre-Isolator, B: Magnetic damper wires, C: Magnetic damper disk, D: Standard filter, E: cabling spider, F: Suspension wire, G: Bottom filter, H: suspension wires of intermediate mass control box, I: intermediate mass control box, J: OSEM position sensor/actuator, K: Intermediate mass, L: whip magnetic damper of mirror recoil mass, M: mirror, N: mirror recoil mass. (**Right**) Illustration of an external support structure supporting the seismic attenuation head of a compact Seismic attenuation chain. Bolted structures are always found to have resonances different and lower in frequency by as much as a factor of 2 than in simulations. This is because in simulations connections between parts are rigid, as welded, while in bolted structures, the connections are always the weakest point. To mitigate this, this support structure is built by two welded frames (**A**,**B**) connected by four over-bolted heavy L-beams (**C**). Over-bolting was common in building bridges before welding was introduced and is important to increase resonant frequencies. Lighter, angled, bolted L-beams (**D**) sandwiching a dissipative rubber layer both stiffen and damp these resonances. The top platform (**E**) supports, through bellows, a short, inverted pendulum pre-attenuator like that of Figure 5, housed in a removable dome (**F**). A much larger and more complex structure would be necessary for the test masses to avoid amplifying low-frequency seismic tilt noise. Such structures would heavily encumber the space around the test masses.

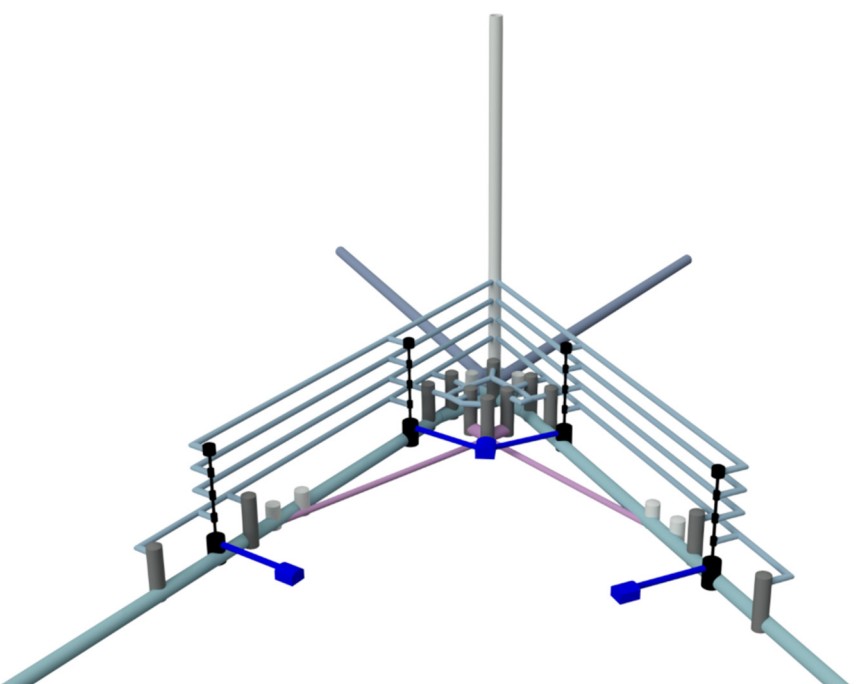

**Figure A2.** Interactive illustration of a possible tunnel configuration with access to the different alcoves and raised bore wells housing the components of the gravitational wave detectors (the interactive file can be downloaded from here and rendered using this viewer). All noisy components, such as the cryogenic chillers, are housed in separate alcoves to confine their noise. The tall vertical well may extend to the surface, providing a required secondary escape route. Note, the main tunnels are illustrated as extending beyond the crossing point. A horizontal tunnel may house the required mode cleaner and vacuum squeezing cavity. The second, angling up, may be the TBM arrival route. If it was convenient to assemble the TBM at the surface, the tunnel would provide the main access route.

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
