# Peer review of "Tunnel Configurations and Seismic Isolation Optimization in Underground Gravitational Wave Detectors"

_applsci, doi:10.3390/app12178827_

Round 1

Author Response

We would like to thank you for the referee report. We appreciated Your comments. Addressing them has improved the quality of the paper. The responses to the referee’s comments are listed below.

The paper proposes an original solution for underground 3rd generation Gravitational Wave detectors which could improve the detector noise suppression and extend the duty cycle while performing structural interventions on some part of the detector itself.

In general, it’s extremely interesting the idea of separating the main optics of the High Frequency and Low Frequency detectors, as well as the possibility of installing in a different cavern the noisy and auxiliary equipment, as long as the costs remain comparable with the original design and the operability is not compromised.

Despite the originality of the proposed solution and the wide interest for this topic, the article is poorly supported by computations/simulations, which would show the strength or weakness of the proposed solution, to the extent that some section results to be extremely qualitative. Also, proper citations are lacking all over the text.

Finally, the paper organization is not very clear. The reader would benefit of a re-shuffling of the paragraphs, as suggested in the comments below.

1 – As a general comment, I would recommend including more citation in this section.

Authors: We included the ET 2020 Technical Design Report and another paper on ET scientific objectives which includes almost everything about ET.

Introduction widely explains the ET observatory background, motivations for underground positioning and xylophone configuration. However, besides lacking sufficient citations, the aim of the study reported in the paper is very poorly illustrated. I would suggest the authors to improve this part, better pointing and describing what are the original aspects which are going to be presented in the paper.

Authors: the original points are pointed out later, in paragraph 2.3, “The proposed tunnel configuration”, in the second part  of chapter 2 “Structure of present and future detectors” while the advantages of the proposed configuration are detailed in in chapter 3.

We considered it reasonable to introduce the new topology after discussing the constraints of the underground location to better present the logics to the reader.

Line 84-87: You say that “. The quieter conditions inducing less Newtonian noise are the main rationale why future gravitational wave detectors aiming to detect gravitational waves below 30-10 Hz will be built underground”. However, this sentence is too generic (future gravitational wave detectors), as it is not the case for Cosmic Explorer, as you point out in lines 90-92.

Authors: we modified “below 30-10 Hz” to “down to 3 Hz” to stress that ET aims to be more sensitive in the low frequency band than CE.  CE that is built on the surface cannot reach this frequency range. We refrained from stressing this fact explicitly since it is a politically sensitive issue.

Line 102: typo: “like close by neutron star” → “like those by neutron star” Section 1.3: add more citations about ET LF and HF configuration.

Authors: Thank you, but we really meant close-by, with meaning that they are close to Earth.  It is rare to have one close-by event, but the large signal to noise signal will allow us to detect the tidal deformation od the neutron stars with sufficient precision to determine the elastic properties of the neutronic structure.  This is described in Sathyaprakash’s document referenced therein.

2 – A general comment about section 2 is that the existent solution and the original one are mixed among the different subparagraphs. Maybe a re-organization of paragraph would better clarify to the reader the difference between the two solutions. Maybe, consider to make a new paragraph with your proposed configuration (perhaps merge with 3?) Also in this section, please add citations in subsections 2.1 and 2.2.

Authors: There are already citations in the two paragraphs that we believe sufficiently illustrate the constraints. We moved the paragraph of the proposed seismic attenuation system at the end of section 2. In this way section 2 is organized such that we discuss before the underground constraints and then we discuss the proposed solution, then we discuss the current suspension systems and after that the proposed suspension system. Thank you for the comment.  

Section 2.1 has the same title as 1.3?

Authors: thanks for noting the second title was wrong it now reads: Underground stability constraints

3 – As a general comment, only review the number of sub-paragraph, and try to standardize it with the previous sections.

Authors: done. 1. became 3.1 and so on, thank you

Line 388: 10-18m/√Hz → 10−18?/√??

Authors: Done

3.4: a 60m long pendulum seems to be a “school case” rather than a real and effective solution. It’s interesting to compare results for configurations different from the official one (17m long pendulum), but if the 60m long pendulum is a realistic proposal, it should be supported by more studies and evaluations, not the least of which the economical one. If these evaluations have been performed, please add a citation. Otherwise, it would be preferable to underline that this solution is a kind of “gedankenexperiment”.

Authors: The 60 m is a length obtained by excavation requirements applied to the topology (5 filters) of the ET current isolation chain configuration.  Figure 2 and 3, obtained with octopus that is the same tool used to design the current ET configuration, illustrate the differences in performance between the two lengths and gain in frequency.

About the economic evaluation, we mentioned that the cost was evaluated to be equivalent to that of the current design.  We would like to stress to the reviewer that the experts making the economic-technical evaluation (referenced) agreed that shorter separations between the levels would require additional reinforcement and add to the cost.

3.6, Line 404: It’s not clear why relocating seismic attenuation and cryogenics away from the main tunnel level would allow to use much smaller vacuum chambers. Could you explain it better?

Authors: thanks for pointing that out, the text was indeed not clear and has been changed in:

“Having relocated the seismic attenuation away from the main tunnel level, much smaller vacuum chambers can be used, including for the test masses.  Similarly, moving the cryogenics in separate excavations, and injecting the heat links higher up in the attenuation chain allow the use of smaller gauge tunnels”

Line 408: “Much cleaner working conditions can be obtained if the operator can work from the side of the vacuum chamber while the optics are continuously protected by a flow of clean air provided from above”. Isn’t it what happens also when the access is from the bottom? “Much cleaner working conditions” with respect to what? Are there measurements or data which can support this sentence?

Authors:  changed to “Much cleaner working conditions than for the test masses of the present detectors can be obtained if the operator can work from the side of the vacuum chamber while the optics are continuously protected by a flow of clean air provided from above.”

3.9: Extremely interesting the “segregating noisy equipment” paragraph!

Authors: thank you

4 – It seems that paragraph 4 is extremely short. Could it be merged into some other paragraph (for instance 3.8)?

Authors: the two paragraphs address to two different and important  issues, in our opinion it would be confusing to merge them.

5 – Line 465: no citation for Virgo magnetic anti-spring system?

thanks for pointing it out; we changed “used in Virgo” to “used in the Virgo upgrades” and added a specific reference.

In effect we referred to the GAS springs that have been used to isolate all Virgo optical bench  upgrades while the original superattenuators remained with the original MAS attenuators.

Magnetic anti springs (both the original MAS and the new GAS filters were designed by one of the authors of this paper) may be used as well, but they are limited by the many internal resonances of the centering mechanism and as a result they are limited to less attenuation. This even without the introduction of the magic wands that in the GAS filters add another order of magnitude of attenuation.

Using GAS filter instead of MAS filters may allow the use of one less level of attenuation.

But this optimization is outside the scope of this paper, that is centered on the tunnel topology.         

Lines 488 and 490: 10-4 → 10−4

Authors: Done

6 – Also this paragraph is extremely short. Could it become the introduction of paragraph 3 or maybe inserted in a “Discussion” paragraph at the end or in the Conclusions?

Authors: we use different paragraphs to address different items, we never set minimum lengths for our paragraphs.

7 – This paragraph seems to be a bit too qualitative. One question about the content is: isn’t the Newtonian noise reconstruction in the frequency band 2-20 Hz (if this is the band, which is not specified either) independent of the detector performance?

Authors: Yes it is.

If not, please cite a paper. If yes, I don’t see why the excavations of boreholes for seismometer installations can’t be performed at the same time as the tunnels and caverns.

Authors: ET will likely start working without a real Newtonian noise cancellation system which will be implemented at a later stage. Indeed, cryogeny will likely not be implemented in the first ET stages, therefore the Newtonian noise will be dominated by the thermal noise. I am afraid there is no paper that says this explicitly.

The description of the sensor installation seems to be very interesting, although a bit imaginative, as it would involve a custom crawler for installing custom triaxial inertial sensors remotely controlled. All of this is extremely interesting, but it seems to be a topic for a dedicated paper, including all the proper studies and citations. What I consider to be important in this paragraph is the idea that boreholes can be placed in the proposed pattern, separated from the main tunnels, in line with the main idea of the paper. However, if you want to add more details, I recommend to support them with more scientific data

Authors: In fact we plan to make it the topic for a dedicated paper.  As you point out, though it is interesting and we think it important to mention it in this paper, as it would become part of the tunnel topology.

In the text we added mention that this will be the topic for a dedicated paper

Reviewer 2 Report

In this manuscripts, the author try to present a configuration that could improve the observatory’s event detection efficiency. In the present model, the gaps in gravitational wave observations and the risk of missing a rare, potentially multiple-messenger event like a close-by supernova or a neutron star merger are eliminated. In addition, distributing the seismic attenuation chain components over multiple tunnel levels allows the use of effectively arbitrarily long seismic attenuation chains that relegate the seismic noise at frequencies farther from the present low-frequency noise budget, thus keeping the door open for future upgrades. Mechanical crowding around the test masses is eliminated allowing the use of smaller vacuum tanks and reduced cross section of excavations, which require less support measures.

The authors have presented few advantages of their presented model. Like, the first and foremost gain from the tunnel topology proposed here is the improvement of the observatory astronomical observation efficiency. The physical separation of the test masses, the recombination optics of different detectors and the support point of the isolation chains, as well as the relegation of noisy equipment in separate and sealed excavations will allow continuous astronomical observation mode in most detectors even while work is done on others for staged installation, maintenance, or upgrades. The second advantage is that arbitrarily long attenuation chains built into the rock environment push the seismic noise farther from the detector noise budget and keep the door open for future sensitivity improvements. Perhaps more importantly, lower frequency attenuation and smaller residual motion reduce the force required for controls and the chance that control noise may end up limiting the detection sensitivity. The cost of boring and of high-quality inertial sensing instruments is relevant, and the quality of sensors continuously improves. It will take years before the Newtonian noise will represent a limit to sensitivity. Therefore it is probably wise to delay implementation of instruments for full-fledged Newtonian noise subtraction. The proposed tunnel topology is suitable for staging without interrupting astronomical observations.

This manuscript is well motivated and well written with reasonably good English. The results will be useful to the researchers in this particular field. Hence I recommend the publication of this work in its present form in your esteemed journal.

Author Response

The Authors woulds like to warmly thank the reviewer for the time spent on this paper and for the kind review.

Reviewer 3 Report

Review Report

Ms. Ref. No.: applsci-applsci-1812710-peer-review-v1, Submitted to MDPI, Journal: “Applied Sciences”

Title:“Tunnel configurations and seismic isolation optimization in underground gravitational wave detectors”

By: F. Amann, F. Badaracco, R. DeSalvo, L. Naticchioni, A. Paoli, L. Paoli, P. Ruggi, S. Selleri

Comments

In this manuscript the authors address a configuration “applicable” to the Einstein Telescope (ET) towards improving drastically the observatory’s event detection efficiency. As is well known, the Einstein telescope is going to be a Gravitational Wave (GW) observatory comprising six nested detectors, Michelson interferometers type gravitational wave detector (of some kilometers long).

The article is, in general, well organized and well written while authors make use of very good English language. Therefore, the paper preserves publication in the J. “Applied Sciences” after improving the following points.

The Einstein Telescope (ET) is “a European third-generation gravitational-wave detector”, based on the measurement of tiny changes in the lengths of connected arms (about 10 kilometers long), caused by a passing gravitational wave. Authors, please, make it explicitly clear in the Introduction.

Authors compare the ET advantages with those of the present surface (L-shaped) detectors Virgo, LIGO and KAGRA detector. Because the operating “Advanced Virgo”, “Advanced LIGO”, are second-generation (interferometric) GW detectors, it is not clear if the comparison is done with the latter class of detectors (inside the text authors use only the term “Advanced LIGO”, P-13, L-506, even though in their referenced list appropriate references are included). Please, make use of the explicit term “advanced” accordingly.

In general, authors do not make normal use of citations inside the text (there exist entire sections and sub-sections without any citation!!). Please, improve throughout the paper accordingly and insert some new references. For example, in Sect. 1.2. put appropriate references/citation discussing the ET sensitivity to events from “close by neutron star inspirals/mergers or supernovae”.

It is beneficial for this paper to include in the reference list the recent “Einstein Telescope Symposium”, Budapest, June 7-8, 2022. Please, if possible, comment on some relevant novelties/conclusions/outlook.

Authors discuss (P-2, L-46) the required ET advantage in “extremely well isolated optical elements” from “undesired vibrations” (e.g. the seismic waves are “undesired” vibrations). Please, discuss them (by involving the necessary references in Sect. 1.2 and elsewhere) in conjunction with “desired weak sources of vibrations” like those of close-by/far neutron star mergers (weak GW events background-signals at the sensitivity-threshold) of ET. Of course, some of the latter sources of vibrations are events detectable by ET.

In summary, the paper is quite interesting and I recommend its acceptance for publication in the Journal “Applied Sciences” of MDPI.

Minor Points

In P-7, L-310: “… capital importance ...” ==> check possible rewording as “… crucial importance …”

I suggest the application of “the \widefigure command” of MDPI in Fig. 1 for better arrangements of the two upper parts of this figure?

Author Response

We would like to thank you for the referee report. We appreciated Your comments. Addressing them has improved the quality of the paper. The responses to the referee’s comments are listed below.

Comments

In this manuscript the authors address a configuration “applicable” to the Einstein Telescope (ET) towards improving drastically the observatory’s event detection efficiency. As is well known, the Einstein telescope is going to be a Gravitational Wave (GW) observatory comprising six nested detectors, Michelson interferometers type gravitational wave detector (of some kilometers long).

The article is, in general, well organized and well written while authors make use of very good English language. Therefore, the paper preserves publication in the J. “Applied Sciences” after improving the following points.

— The Einstein Telescope (ET) is “a European third-generation gravitational-wave detector”, based on the measurement of tiny changes in the lengths of connected arms (about 10 kilometers long), caused by a passing gravitational wave. Authors, please, make it explicitly clear in the Introduction.

Authors: We have added a few lines in section 1.1

— Authors compare the ET advantages with those of the present surface (L-shaped) detectors Virgo, LIGO and KAGRA detector. Because the operating “Advanced Virgo”, “Advanced LIGO”, are second-generation (interferometric) GW detectors, it is not clear if the comparison is done with the latter class of detectors (inside the text authors use only the term “Advanced LIGO”, P-13, L-506, even though in their referenced list appropriate references are included). Please, make use of the explicit term “advanced” accordingly.

Authors: We have added "Current" in line 112, where we mention them for the first time

— In general, authors do not make normal use of citations inside the text (there exist entire sections and sub-sections without any citation!!). Please, improve throughout the paper accordingly and insert some new references. For example, in Sect. 1.2. put appropriate references/citation discussing the ET sensitivity to events from “close by neutron star inspirals/mergers or supernovae”.

— It is beneficial for this paper to include in the reference list the recent “Einstein Telescope Symposium”, Budapest, June 7-8, 2022. Please, if possible, comment on some relevant novelties/conclusions/outlook.

Authors: We some references, instead of the ET Symposium reference we preferred to add the updated Technical Design Report (2020) which contains all the useful information about ET. In some sections we thought that references were not necessary either because useful referencing had already been done earlier in the text, or because the sections discuss uniquely features proposed in this work, and therefore do not require referencing.

— Authors discuss (P-2, L-46) the required ET advantage in “extremely well isolated optical elements” from “undesired vibrations” (e.g. the seismic waves are “undesired” vibrations). Please, discuss them (by involving the necessary references in Sect. 1.2 and elsewhere) in conjunction with “desired weak sources of vibrations” like those of close-by/far neutron star mergers (weak GW events background-signals at the sensitivity-threshold) of ET. Of course, some of the latter sources of vibrations are events detectable by ET.

Authors: We added “seismic induced” in line 46. About all the astrophysical stochastic background that might create a confusion noise in ET it is something out of the scope of the present paper, this is why we didn't mention it.

In summary, the paper is quite interesting and I recommend its acceptance for publication in the Journal “Applied Sciences” of MDPI.

Minor Points

— In P-7, L-310: “… capital importance ...” ==> check possible rewording as “… crucial importance …”

Authors: done

— I suggest the application of “the \widefigure command” of MDPI in Fig. 1 for better arrangements of the two upper parts of this figure?

Authors: We made it a bit larger

Reviewer 4 Report

The study reports a tunnel topology approach for the improvement of the gravitational wave observation efficiency. The authors are requested to address the following concerns:

1)    Although the paper has some merit, it is presented in a long-winded way. The authors should condense the paper, writing in a concise and clear way that is appropriate for a technical paper rather than a tutorial in a text book. Note that some of the contents are well-known facts in the GW community. the authors should state clearly where is the novelty of their approach, without this I can’t recommend this work for publication.

2)    Only passive seismic vibration isolation systems are considered/reviewed in the paper. Does this imply that the presented seismic isolation systems will be implemented for ET? As far as I know, there are on-going projects investigating the potential of combing the aLIGO’s active platforms and aVirgo’s passive platforms for improving the low-frequency seismic isolation. The low-frequency seismic noise is critical as they might affect the ET’s sensitivity through up-conversion effects. The authors are asked to comment on this.

Author Response

We would like to thank you for the referee report. We appreciated Your comments. The responses to the referee’s comments are listed below.

Comments and Suggestions for Authors

The study reports a tunnel topology approach for the improvement of the gravitational wave observation efficiency. The authors are requested to address the following concerns:

1)    Although the paper has some merit, it is presented in a long-winded way. The authors should condense the paper, writing in a concise and clear way that is appropriate for a technical paper rather than a tutorial in a text book. Note that some of the contents are well-known facts in the GW community. the authors should state clearly where is the novelty of their approach, without this I can’t recommend this work for publication.

The Authors found that the present comment is in disagreement with the other 3 reviewers:

Rev 1: “English language and style are fine/minor spell check required ”

Rev. 2: “This manuscript is well motivated and well written with reasonably good English. The results will be useful to the researchers in this particular field. Hence I recommend the publication of this work in its present form in your esteemed journal.”

Rev. 3: “The article is, in general, well organized and well written while authors make use of very good English language. Therefore, the paper preserves publication in the J. “Applied Sciences” after improving the following points. . . .    In summary, the paper is quite interesting and I recommend its acceptance for publication in the Journal “Applied Sciences” of MDPI.”

We believe that shortening the paper may make it less readable for the not-expert reader and we would prefer (unless specifically required) to leave it as is, of course after the implementation of all the other suggested improvements that we have done.

2)    Only passive seismic vibration isolation systems are considered/reviewed in the paper. Does this imply that the presented seismic isolation systems will be implemented for ET? As far as I know, there are on-going projects investigating the potential of combing the aLIGO’s active platforms and aVirgo’s passive platforms for improving the low-frequency seismic isolation. The low-frequency seismic noise is critical as they might affect the ET’s sensitivity through up-conversion effects. The authors are asked to comment on this

Authors: We clearly stated that this is a proposal to be evaluated, in order to increase the effectiveness of ET as an astronomical observatory (chapter 1.2), it is up to the ET collaboration to accept in part or in toto, or refuse the ideas presented here. This may be an alternative to other ongoing projects or be complemented by active components that can be implemented in the proposed topology.

Two reviewers stated that “the methods not adequately described” and should be improved.

As mentioned above, the paper is fundamentally a suggestion to change the topology of ET to improve its effectiveness, as well grasped for example by Rev. 1

“The paper proposes an original solution for underground 3rd generation Gravitational Wave detectors which could improve the detector noise suppression and extend the duty cycle while performing structural interventions on some part of the detector itself.  In general, it’s extremely interesting the idea of separating the main optics of the High Frequency and Low Frequency detectors, as well as the possibility of installing in a different cavern the noisy and auxiliary equipment, as long as the costs remain comparable with the original design and the operability is not compromised.”

Besides this there are no specific methods that need to be applied.  The methods that we chose to make a coherent story are already specifically described elsewhere, to which references are provided.

Round 2

Reviewer 4 Report

The paper is improved by considering the comments provided by the reviewers. However, this reviewer still has an issue with the paper, which is principally with the style of the writing. Many introductions can be tighten up as long as the reader can get the idea from the references. I will not give an exhaustive list of what I mean as that would impose my style. However, one example is:

The contents that presented in the subsection 5.1 Vertical attenuation filters mainly review the existing systems, which is not appropriate to be organized in the main content of the paper. 

This is meant to give an idea as to what should be improved with the style of the presentation. I remain the opinion that if the manuscript is not revised to a high standard.  

Author Response

The authors would like to thank the reviewer for the comments. However The authors are sorry that this referee still disagree with the other three. While many of the authors of this article may be less experienced, at least three of them are senior and reviewers of scientific journals. We all agree that although the English may not be at the level of a bestseller novel, all the scientific points that we wanted to make are clearly understandable  and therefore it is adequate for publication. The style is to explain, even to non-specialists, with simple arguments and examples, how a system taking advantage of the proposed topology may work. This is the reason subsections like the 5.1 mentioned are in our opinion useful.